# Myxedema in Both Hyperthyroidism and Hypothyroidism: A Hormetic Response?

**DOI:** 10.3390/ijms25189957

**Published:** 2024-09-15

**Authors:** Salvatore Sciacchitano, Angela Napoli, Monica Rocco, Claudia De Vitis, Rita Mancini

**Affiliations:** 1Department of Life Sciences, Health and Health Professions, Link Campus University, Via del Casale di San Pio V, 00165 Rome, Italy; 2Departmental Faculty of Medicine, Unicamillus International University of Health and Medical Sciences, Via di Sant’Alessandro 8, 00131 Rome, Italy; angela.napoli@unicamillus.org; 3Department of Clinical and Surgical Translational Medicine, Sapienza University, Via di Grottarossa 1035/1039, 00189 Rome, Italy; monica.rocco@uniroma1.it; 4Department of Clinical and Molecular Medicine, Sapienza University, Via di Grottarossa 1035/1039, 00189 Rome, Italy; claudia.devitis@uniroma1.it (C.D.V.); rita.mancini@uniroma1.it (R.M.)

**Keywords:** myxedema, thyroid function, bioelectrical impedance analysis (BIA), hormesis

## Abstract

Myxedema is a potentially life-threatening condition typically observed in severe hypothyroidism. However, localized or diffuse myxedema is also observed in hyperthyroidism. The exact cause and mechanism of this paradoxical situation is not clear. We report here the analysis of body fluid distribution by bioelectrical impedance analysis (BIA) in 103 thyroid patients, subdivided according to their functional status. All BIA parameters measured in subclinical thyroid dysfunctions did not significantly differ from those observed in euthyroid controls. On the contrary, they were clearly altered in the two extreme, opposite conditions of thyroid dysfunctions, namely overt hyperthyroidism and severe hypothyroidism, indicating the occurrence of a typical hormetic condition. Surprisingly, differences in BIA parameters related to fluid body composition were even more evident in hyperthyroidism than in hypothyroidism. A hormetic response to thyroid hormone (TH)s was previously reported to explain the paradoxical, biphasic, time- and dose-dependent effects on other conditions. Our results indicate that myxedema, observed in both hypothyroid and hyperthyroid conditions, represents another example of a hormetic-type response to THs. BIA offers no additional valuable information in evaluating fluid body composition in subclinical thyroid dysfunctions, but it represents a valuable method to analyze and monitor body fluid composition and distribution in overt and severe thyroid dysfunctions.

## 1. Introduction

Myxedema is defined as a specific form of cutaneous and dermal edema secondary to the increased deposition of connective tissue components. In particular, this condition is characterized by the accumulation of protein-mucopolysaccharide complexes that bind water and are responsible for the occurrence of non-pitting, boggy edema. The occurrence of myxedema in thyroid disorders has been known for many years. In 1981, the English physician G. R. Murray successfully cured a 46-year-old woman suffering from myxedema due to hypothyroidism by administering her an intravenous extract prepared from a sheep’s thyroid [1]. However, the evidence of myxedema has also been reported in patients affected by hyperthyroidism. Late in 1965, Frederick A.J. Kingery reported, with surprise, a case of myxedema in a hyperthyroid individual who almost invariably had exophthalmos [2]. He noted that the localized myxedema, observed in hyperthyroid patients, was the same as that reported in those with overt hypothyroidism, caused by a medical or surgical suppression of thyroid activity. The accumulation of mucin in the dermal connective tissue, under the skin, was demonstrated by histochemistry using staining specific for acid mucopolysaccharides, such as Alcian blue, methylene blue, or toluidine blue [3]. However, the exact pathogenic mechanism of myxedema occurring in thyroid disorders is not clear yet and the reason why it can be observed in both hypothyroid and hyperthyroid patients is not known either.

BIA represents an efficient, low-cost, and easy-to-apply method for evaluating body composition in health and disease [4]. In addition, it is particularly suitable to assess total body water volume [5] and to evaluate fluid composition and distribution in the body as well as cell membrane integrity [6,7]. It is based on the principle that various body components offer a different resistance to the passage of an electrical current [8]. In particular, it assumes that the resistance to a determined electrical current is inversely proportional to the distribution of TBW and electrolytes. In general, lean tissues, composed of muscle mass, bone mass, electrolytes, and water, are considered high conductors of electrical currents. Therefore, they offer low resistance to the passage of the electrical current. Conversely, fat, bone, and skin, composed of a smaller quantity of fluids and electrolytes, show low conductivity and therefore, they offer high electrical resistance. The raw parameters of BIA are Xc and Rz. These data are used to estimate the amount of fat mass (FM), fat-free mass (FFM), total body water (TBW), extracellular water content (ECW), and intracellular water content (ICW). It is noteworthy that the hydration of fat-free body mass, expressed as the ratio FFM/TBW, is remarkably stable at approximately 0.73, and such a ratio represents a cornerstone in the body-composition research field [9]. It usually increases in cases of accumulation of large volumes of fluid. Another relevant parameter is represented by the Na*_e_*:K*_e_* ratio. Considering the importance of the sodium/potassium pump in the energetic metabolism of the cells, this ratio has been proposed as an objective marker of nutritional status and an excellent indicator of an adverse clinical outcome. It has been used to identify malnourished patients at risk of dying [10]. Total exchangeable sodium (Na*_e_*), total exchangeable potassium (K*_e_*), and TBW are the major determinants of the plasma water sodium concentration [11]. The phase angle (PhA) represents an indicator of cell membrane integrity, of body cell mass, and of the distribution of water within and outside the cell membrane. It is a valuable tool for the assessment of nutritional status of hospitalized patients with different co-morbidities, including hemodialysis patients [12,13,14], hospitalized children [15], geriatric patients [16], and cancer [17,18,19,20,21]. PhA is a valid predictor of mortality in elderly patients [22], as well as in many different diseases [23]. PhA is usually higher in healthy conditions, and it is lower in the occurrence of various diseases as well as in aging.

To examine the characteristics of myxedema in both hyperthyroidism and hypothyroidism, we analyzed body fluid composition by means of a single-frequency bioelectrical impedance analysis (SF-BIA) in patients, affected by thyroid dysfunction and subdivided according to the levels of thyroid hormones (TH)s in serum. Our data indicate that a correlation exists between THs and some parameters of BIA, especially with reactance, hydration, sodium/potassium exchange ratio, and phase angle. Such correlations follow a U-shaped or an inverted U-shaped curve, suggesting the occurrence of a hormetic-type response in which both marked increase and marked decrease in thyroid hormones serum levels are associated with fluid abnormalities at the periphery, with the sodium/potassium pump probably involved in such effect.

The terms hormesis and hormones share a common Greek origin. The original Greek name hórmēsis describe something that is in rapid motion. In the case of hormones, the term is used to describe a class of signaling molecules setting in motion because they are synthesized and secreted by endocrine organs that are distant to tissues where they exert their complex biological processes to regulate physiology and behavior and to maintain homeostasis. Hormesis is emerging as a new concept with universal applicability and a high degree of generalizability in many different fields, including environmental science, biology, toxicology, and medicine [24,25,26]. The hormetic response in medicine represents an adaptive response of cells, tissues, organs, and organisms to changes in environmental conditions and to exposure to several different stressors, such as metals, vitamins, macronutrients, micronutrients, drugs, and virtually, to every physical, chemical, and potentially toxic agent. The hormetic response depends on the dose and time of exposure to stressors. The following are examples of physical stressors: (i) ionizing radiation, (ii) energy and caloric deficiency, (iii) temperature, (iv) low nitrogen, and (v) low oxygen. Chemical stressors are represented by the following: (i) ROS (reactive oxygen species), (ii) herbicide and phytochemicals, (iii) curcumin, (iv) antibiotics, and (v) trace elements or heavy metals. Among the biological stressors there are the following: (i) viruses, (ii) bacteria, and (iii) parasites.

The results of our study indicate that the occurrence of myxedema in both severe hypothyroidism and overt hyperthyroidism could represent another example of hormetic response with respect to the action of thyroid hormones on fluid and electrolyte balance.

## 2. Results

The resistance-versus-reactance (R-Xc) graphs of vector BIA in the different groups of patients, subdivided according to the thyroid function, are reported in Figure 1. No significant differences were observed between euthyroidism (Figure 1A) and both subclinical hypothyroidism (Figure 1D) and subclinical hyperthyroidism (Figure 1B). We observed that both our overt hyperthyroid (Figure 1C) and severe hypothyroid patients (Figure 1F) showed a displacement of vectors toward the downward direction of the semi-major axis of the ellipse, indicating a high prevalence of fluid retention, in the form of edema in these patients, compared to our control patients that were almost all included inside the 95% and 75% tolerance ellipses (Figure 1A). Thyroid patients with overt hyperthyroidism showed a more profound effect on edema when compared to overt (Figure 1E) and severe hypothyroidism (Figure 1F), and this effect was associated with a reduction in muscular mass (MM), with a consequent shift toward the right of the semi-minor axis of the ellipse, particularly visible in two of them with hyperthyroidism lasting for more than three months (Figure 1C, black arrows).

The analysis of the different BIA parameters with respect to the thyroid functional status is reported in Table 1. It reveals some interesting findings. Many of the parameters related to fluid balance show a peculiar behavior with medium values that appear to be similarly altered in the two opposite conditions of overt hyperthyroidism and severe hypothyroidism. In particular, reactance (Xc) and phase angle (PhA) were reduced in these two conditions, compared to the euthyroid group of patients. Conversely, hydration, measured as TBW/FFM, and the sodium/potassium exchangeable ratio (NA*_e_*:K*_e_*) were both higher when compared to euthyroid condition. A similar effect of two opposite thyroid functional statuses resembles those reported for micronutrients and macronutrients and known as the hormetic effect [25]. In agreement with the hormetic response, the relationship between thyroid hormonal status and many BIA parameters, especially those related to fluid balance, do not appear to follow a linear distribution. The relationship between thyroid hormonal status and BIA parameters of hydration, reactance, phase angle, and the Na/K exchangeable ratio are, in fact, better represented as a U-shaped or an inverted U-shaped (bell-shaped) curve (Figure 2). This situation is observed in many conditions in which duality in function suggests the occurrence of hormesis [27].

The opposite extreme conditions of overt hyperthyroidism and severe hypothyroidism are characterized by a similar alteration of the BIA parameters. The hydration, measured as the ratio between FFM and TBW, and the Na/K exchangeable ratio are both increased in these conditions as compared with the euthyroid control group of patients (Figure 2A,C). Conversely, the phase angle (PhA) and the reactance (Xc) are both reduced compared to the euthyroid control group of patients (Figure 2B,D). In almost all cases the difference is statistically different and more evident in the overt hyperthyroid group of patients in comparison with the euthyroid controls, indicating the occurrence of a more severe fluid imbalance in overt hyperthyroidism than in severe hypothyroidism.

## 3. Discussion

All BIA parameters measured in our patients appear to follow a peculiar distribution in relation to TH status. In particular, PhA, ECW, Na*_e_*:K*_e_*, and hydration, measured as total body water/fat-free mass (TBW/FFM), appear to be altered in both hypothyroid and hyperthyroid patients, and when patients are subdivided according to the severity of the thyroid dysfunction, such parameters are distributed in a U-shaped curve or in an inverted U-shaped curve. The correlation between thyroid hormone serum levels and these BIA parameters could be represented as a nonmonotonic or curvilinear relationship. In other words, these results indicate that there is an improvement in these BIA parameters when serum thyroid hormones fall in the range of intermediate levels, while the exposure to both too-low and too-high serum thyroid hormone levels is responsible for the occurrence of imbalance in fluid distribution that may ultimately lead to the development of myxedema. The inverted U-shaped curve of the PhA, observed in our patients with thyroid dysfunctions, indicates that both severe hypothyroidism and overt hyperthyroidism are responsible for a reduction in the PhA, suggesting a remarkable damage in cell membrane, with consequent occurrence of edema in these two conditions. Reactance is a measure of opposition to the flow of current caused by the capacitance produced by the cell membrane. It gauges cells’ ability to store energy. In general, high levels of reactance indicate that cells are healthy, with intact cellular membranes, can easily store energy, and hold the electrical energy charge longer. Conversely, low reactance indicates that cells are less efficient in storing energy and in maintaining electrical energy charge. We observed such conditions with low reactance in both severe hypothyroidism and overt hyperthyroidism that again, appear to have similar negative effects with respect to the maintenance of cell integrity. The FFM is subdivided in BCM, which represents the metabolically active, oxygen-consuming part of the body and is composed of muscles and internal organs, and in ECM, which reflects the more inactive constituents involved in transport and support, such as extracellular fluids and the skeleton. In healthy subjects the ECM/BCM ratio was reported to be near 1 [28]. In our patients, a higher ECM value was found in both overt hyperthyroidism and severe hypothyroidism, compared to euthyroid control patients, reflecting the imbalance in fluid distribution with extracellular accumulation of hygroscopically active glycosaminoglycanes, typically found in myxedema in both conditions.

BIA was previously used to assess body composition in correlation with thyroid function in healthy subjects and in those with different thyroid dysfunctions. In healthy subjects, body resistance, measured as an index of fat-free tissues (which are responsible for 95% of basal energy expenditure) was considered a better indicator of thyroid function than anthropometry [29]. In patients with impaired thyroid function, significant differences were reported in the bioelectrical resistance, reactance, and in the PhA [30]. However, results of this study were probably altered by the presence of relevant differences in the anthropometric data of the various groups of patients included. In this regard, we should consider that the results of the BIA are influenced by many factors, including age, sex, and BMI. In our patients, there was a clear predominance of females, and the F/M ratio was higher in the euthyroid control group (7.1:1). This difference was much higher in patients with both hyperthyroidism (10.5:1) and hypothyroidism (9:1). Our patients with thyroid dysfunction; however, did not differ with respect to their age and BMI when compared to the euthyroid control group (Table 2).

We previously reported that COVID-19 patients frequently presented a reduction in the free triiodothyronine (FT3) serum levels, a condition known as low T3 syndrome or nonthyroidal illness syndrome (NTIS) [31]. Currently, there is no clear evidence in favor of treatment with either triiodothyronine (T3), thyroxine (T4), or both in such conditions [32]. In these patients, and especially in those with severe COVID-19 and admitted to ICU, we demonstrated the usefulness of BIA to evaluate body composition and, in particular, the fluid distribution responsible for the occurrence of myxedema and generalized anasarca [33]. We found similarities between the hydroelectrolytic alterations observed in these patients and those observed in one single patient with myxedema. Such similarities included the reduction in the PhA and ICW and the increase in hydration, measured as a TBW/FFM ratio, in ECW and in Na*_e_*:K*_e_*. Based on these observations and on a quantification of the expression of the two major genes encoding for the sodium/potassium pump, we suggested that the effect of thyroid hormones on fluid balance could be exerted through a direct effect, at the transcriptional level, on these genes, and we proposed BIA as a useful method to monitor hydroelectrolytic balance at the periphery in patients with NTIS. Myxedema; however, is not associated only with hypothyroidism. As previously reported, either localized or more diffuse edema can be understood also in hyperthyroid patients.

In the present study, we extended our previous observations to patients affected by different thyroid dysfunctions. We report here the results of BIA in a group of 103 patients with different thyroid hormonal dysfunctions. We aimed to find useful insights on the reason why myxedema can be detected in both thyroid dysfunctions. Surprisingly, we found that the fluid distribution in the bodies of our patients shows similar results in the two extremities on the spectrum of thyroid hormone function. The most severe forms of hyperthyroidism and of hypothyroidism are, in fact, characterized by similar alterations of some of the BIA parameters and, in particular, of those related to Xc, PhA, TBW/FFM, and the Na*_e_*:K*_e_*. Our results indicate that these BIA parameters follow a U-shaped or an inverted U-shaped curve, with respect to thyroid hormonal function, and confirm the clinical observation that myxedema occurs when thyroid hormones in serum are either too low or too high. These results resemble the typical situation observed with nutrients and minerals, known as the hormetic response. Such an effect was not reported in a previous study in which only patients affected by subclinical hypothyroidism were analyzed [34]. The results of this study are in agreement with our observation regarding patients with subclinical hyperthyroidism and hypothyroidism, in which BIA parameters were similar to those observed in patients with euthyroid conditions. The major changes in BIA parameters are visible only when patients showed clearly altered thyroid hormones in serum. Therefore, the measurement of BIA parameters is useful only in patients with overt hyperthyroidism and severe hypothyroidism, while the influence of thyroid hormones in subclinical thyroid dysfunction seems to be not relevant.

Despite the similarities in the name and in functions between hormesis and hormones, there are only a few articles in the literature concerning any correlation between them. Nevertheless, it is clear to any endocrinologist that working with hormones essentially means to maintain the concentrations of hormones in blood circulation and at the cellular level in a physiological range. Hormones are usually secreted at very low concentrations, in the order of micro or picomolar range. Yet, their activity is so potent that they can regulate nutrients and substances present in a concentration often more than one- million folds higher in the blood. Any subtle variation in their concentration is responsible for a dramatic effect in the regulated molecules or functions. The concept that too little and too many hormones can be harmful for the cells, tissues, organs, and for the entire organism is, therefore, very familiar to endocrinologists. However, the fact that the behavior of hormones is closely related to the hormetic effect is not well understood and considered yet. The biphasic dose–effect or time–effect relationship for endocrine factors and its impact on health and chronic disease has been recently reviewed [35]. There are many examples of endocrine factors that show opposite effects depending on their levels, which may fall within the broad definition of hormesis. In the case of THs, it has been reported that hyperthyroidism increases the risk of coronary heart diseases, pulmonary hypertension, and atrial fibrillation, while hypothyroidism is responsible for left ventricular diastolic dysfunction and may increase carotid intima-media thickness [36]. Another example of such a biphasic dose–effect of THs is related to metabolism. THs are known to increase catabolism and energy expenditure, but T3 levels have been reported to be positively correlated with unfavorable metabolic parameters in some populations [37,38]. Finally, a hormetic response to THs has also been suggested to explain the paradoxical effects of T3 on calorigenic conditions involving mitochondrial O_2_ consumption and oxidative stress [39]. In the present study, we report another example of this biphasic and dose-dependent response to THs. We demonstrate that too much as well as too little THs in the serum of our patients is responsible for similar adverse effects on fluid distribution in the body. The BIA parameters of our patients, and especially those related to the hydration of FFM, confirmed the initial clinical observation of Frederick A.J. Kingery that the localized myxedema, observed in hyperthyroid patients, was the same as that reported in those with overt hypothyroidism [2]. According to our results, in both severe forms of hypothyroidism and overt hyperthyroidism, there is, in fact, a similar imbalance in fluid distribution. Such imbalance in fluid distribution, therefore, is not uniquely observed in patients affected by hypothyroidism, but it is also evident in patients with overt hyperthyroidism, where it is even more intense. The exact mechanism of this imbalance in fluid distribution is not clear yet. In our previous experience with COVID-19 patients with non-thyroidal illness syndrome (NTIS), we observed that the Na*_e_*:K*_e_* exchangeable ratio was inversely correlated with FT3 values [33]. Total exchangeable sodium (Na*e*), total exchangeable potassium (K*e*), and TBW are the major determinants of the plasma water sodium concentration [11] and the Na/K pump is a well-known target of T3 action [40,41]. The main function of the Na/K pump is to create and maintain an electrochemical gradient across the plasma membrane and is critical for the resting membrane potential, electrical activity of muscle and nerve, Na+-coupled transport, transepithelial transport, nutrient uptake, osmotic balance, and cell volume regulation [42,43]. Its activity ensures the exit of sodium ions out of the cell in exchange for potassium ions entering the cell. Based on such previous observations and considering the relationship between THs and the observed effect on the Na*_e_*:K*_e_* exchangeable ratio, we may speculate that the regulation of the expression and activity of this pump may play a central role in the occurrence of such a biphasic dose-dependent response. Considering the relevance of the Na/K pump in mitochondrial function, it is likely that THs act as hormonal stressor agents, eliciting a hormetic effect at the mitochondrial level, the so-called mitohormesis [44]. In this regard, by analyzing COVID-19 patients with NTIS, we previously observed an opposite effect of thyroid hormones with respect to oxidative stress and mitochondrial respiration in immune-circulating cells of these patients, indicating that the effects of thyroid hormones are relevant for mitochondrial function [45]. We have analyzed changes in fluid distribution using the BIA 101 analyzer apparatus that allows total-body changes. The use of other BIA analyzers, able to perform regional measurements, could be useful in evaluating local alterations of fluid distribution, such as the pretibial myxedema observed in patients suffering from Graves’ disease.

The use of other techniques for edema detection and quantitation, such as those based on diffusion-weighted MRI, usually applied to detect and measure edema in brain injuries [46,47], in the heart affected by acute myocardial infarction [48], as well as in other clinical conditions [49], may furnish more accurate results. In this regard, it is noteworthy to mention the application of this technique to the analysis of orbital lesions, characterized by edema and swelling, typically observed in Graves’ ophthalmopathy [50,51,52]. Such techniques have been widely applied to the analysis of thyroid glands affected by several different diseases and the evaluation of suspicious thyroid nodules. However, we did not find in the literature any examples of an analysis of the edema associated with thyroid dysfunctions. In any case, the application of the diffusion-weighted MRI to the analysis of edema caused by thyroid dysfunctions appears less feasible, mainly because of their cost, usually more than USD 13,000 [53], and the technical difficulties that require trained and expert staff. On the contrary, BIA represents a very low-cost and easy-to-apply method suitable for this purpose.

## 4. Materials and Methods

### 4.1. Patient’s Recruitment

Following the previous study in which we examined body composition in COVID-19 patients and in control patients with thyroid diseases and a different thyroid status [34], we extended our analysis by focusing on the control group of thyroid patients and by also incorporating patients with overt and severe thyroid dysfunction that were not included in the previous study. We enrolled a total of 103 patients affected by various thyroid dysfunctions and were admitted to our outpatient clinic. The demographic and clinical characteristics of our patients are reported in Table 2.

### 4.2. Measurement of Serum Thyroid Hormones

The serum levels of thyroid hormones (TH)s were measured using a chemiluminescent microparticle immunoassay (CMIA), an immunoassay analyzer (ARCHITECT i1000SR, Abbott Lab., Abbott Park, IL, USA), and specific, dedicated diagnostic kits (ARCHITECT Free T3, FT4, and TSH assay, Abbott Lab., Abbott Park, IL, USA). Conventional reference intervals were 3–6 pg/mL for FT3, 0.7–2.2 ng/mL for FT4 and 0.5–4 μU/mL for TSH, respectively. In particular, a total of 57 patients were in euthyroid conditions, 21 were hyperthyroid showing either overt (TSH suppressed and elevated THs) or subclinical (TSH < 0.2 μIU/mL and normal THs) hyperthyroidism, and 25 were classified as hypothyroid, either subclinical (TSH > 4 μIU/mL, but <10 μIU/mL and normal THs), overt (TSH >10 μIU/mL, but <100 μIU/mL and low THs), or severe (TSH > 100 μUI/mL and low THs). The hyperthyroid, euthyroid, and hypothyroid patients were similar with respect to their age and BMI. Females were slightly more prevalent among both hyperthyroid and hypothyroid patients when compared to the control group of euthyroid patients. Patient characteristics as well as mean values of serum thyroid hormone levels in the different groups of patients are reported in Table 2.

We then subdivided patients in the hyperthyroid group into those with overt hyperthyroidism (n = 4) and those with subclinical hyperthyroidism (n = 17). The patients in the hypothyroid group were subdivided into three groups, namely those presenting subclinical hypothyroidism (n = 12), overt hypothyroidism (n = 7), and those with severe hypothyroidism (n = 6).

### 4.3. BIA

We analyzed the body composition of our patients using a single-frequency bioelectrical impedance analysis (SF-BIA), namely the BIA 101 analyzer (Akern Srl, Pontassieve, Firenze, Italy), as previously reported [34]. Briefly, the measurements were taken while the subject was supine in bed for at least 5 min, to allow a compensation of fluid level in the body. Measurements were performed by exposing the body to the passage of a painless, low-intensity, imperceptible electrical current (500 to 800 μA) at a fixed (≈50 kHz) frequency, applied through cables connected to electrodes (Biatrodes electrodes, Jatreia, Pescantina, Verona, Italy), placed in contact with the skin of the hand, wrist, foot, and ankle of the nondominant side of the body. The source (distal) electrodes, located on the hand and foot introduce the electrical current and the sensor (proximal) electrodes, located on the wrist and ankle, captured the fall in voltage provoked by impedance. The stature of the patients was measured to the nearest 0.5 cm, and weight was measured to the nearest 0.1 kg. BIA was performed at the time of blood withdrawal for the measurement of serum values of thyroid hormones. We measured the following parameters: (i) the resistance (Rz), (ii) the reactance (Xc), and (iii) the arc tangent of the proportion between these two components, known as the phase angle (PhA). Based on these parameters, we obtained estimates of many BIA parameters, including the body cell mass (BCM), the total body water (TBW), the extracellular water (ECW), the intracellular water (ICW), the Na:K exchange rate (Na*_e_*:K*_e_*), the fat-free mass (FFM), and the fat mass (FM). In addition, we calculated the hydration and nutrition state, using the following formulas: hydration = TBW/FFM and nutrition = mg/24 h/htm [53]. BIA was performed using the Bodygram PRO software (vers. 3.0, Jatreia, Pescantina, Verona, Italy).

### 4.4. Statistical Analysis

Statistical comparisons were performed between groups using T-tests to examine categorical data (Euthyroid, Overt Hyper, and Severe Hypo). Results were expressed as means ± SEM. A *p*-value of 0.05, 0.005, 0.0005, 0.00005 or less was used as a criterion to indicate statistical significance. NS = not significant. Data were statistically analyzed using the GraphPad Prism software (vers. GP9-2273399-RKSP-6761C, GraphPad Software, San Diego, CA, USA).

## 5. Conclusions

In conclusion, we report our experience regarding the occurrence of myxedema, analyzed by a BIA in thyroid patients with different TH status. BIA parameters, especially those related to fluid distribution in the body, showed similar alterations in patients with severe hypothyroidism and overt hyperthyroidism, suggesting that body fluid distribution in response to THs follows the typical pattern of a hormetic response. BIA is a fast, safe, portable, low-cost, and easy-to-use tool for evaluating body fluid composition and it represents a valuable tool in the identification and monitoring of hydroelectrolytic changes occurring in patients with myxedema due to severe thyroid dysfunction, both in hypothyroidism and hyperthyroidism.

## Figures and Tables

**Figure 1 ijms-25-09957-f001:**
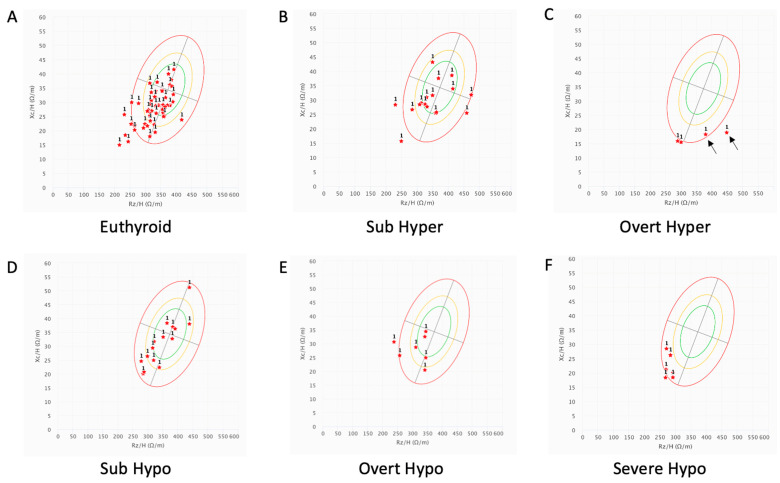
R-Xc (resistance-versus-reactance) graphs of bioelectrical impedance vector analysis data of patients with thyroid dysfunctions, compared to those in euthyroid condition. Red stars indicate individual measurement at admission. The 95% (green line), 75% (yellow line), and 50% (red line) tolerance ellipses are shown. The semi-minor axis of the ellipse indicates cellularity (above, to the left, indicate more body cell mass, and below, to the right, less body cell mass), while the semi-major axis of the ellipse indicates hydration (dehydration towards the upper pole, hyperhydration with apparent edema toward the lower pole). (**A**) euthyroid control patients (n = 57); (**B**) patients with subclinical hyperthyroidism (n = 17); (**C**) patients with overt hyperthyroidism (n = 4); black arrows indicate two cases with overt hyperthyroidism lasting for more than three months; (**D**) patients with subclinical hypothyroidism (n = 12); (**E**) patients with overt hypothyroidism (n = 7); (**F**) patients with severe hypothyroidism (n = 6). Number 1 indicates that BIA parameters have been obtained at initial evaluation.

**Figure 2 ijms-25-09957-f002:**
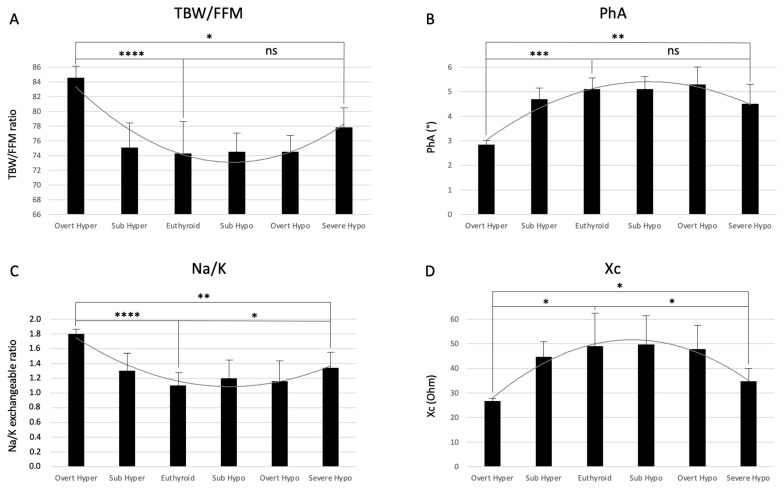
Bar graphs of BIA parameters measured in patients subdivided according to their thyroid functional status. (**A**) TBW/FFM ratio; (**B**) Phase angle; (**C**) Na/K exchangeable ratio; (**D**) Reactance (Xc). The fitted polynomial curves were plotted to better visualize the non-linear relationship between BIA parameters and thyroid functional status. Statistical significance is expressed as follows: ns = not significant, * *p*-value ≤ 0.05, ** *p*-value ≤ 0.005, *** *p*-value ≤ 0.0005, **** *p*-value ≤ 0.00005.

**Table 1 ijms-25-09957-t001:** BIA parameters and thyroid hormone functional status.

*Thyroid Function*	Hyperthyroidism		Euthyroidism	Hypothyroidism		
	Overt	Subclinical		Subclinical	Overt	Severe
FT3 (pg/mL)	14.6	3.2	2.9	2.7	1.9	1.9
FT4 (ng/mL)	12.6	1.4	1.2	1.1	1.4	0.4
TSH (mU/mL)	0.0	0.1	1.7	6.3	22.1	130.9
** *BIA Parameters* **						
Resistance (Rz) 50 kHz (Ohm)	549.5	561.5	540.3	552.3	522.57	440.5
Reactance (Xc) 50 kHz (Ohm)	26.7	44.7	49.0	49.6	47.71	34.8
Muscular Mass (MM)	17.1	26.5	29.6	26.5	32.49	28.4
Muscular Mass % (MM%)	27.3	40.3	42.2	44.6	43.01	40.7
Fat Mass (FM)	24.0	20.3	23.2	16.5	24.54	20.7
Fat Mass % (FM%)	38.2	30.4	30.9	26.3	29.97	29.3
Fat-Free Mass (FFM)	38.4	45.2	48.6	43.8	51.81	49.5
Fat-Free Mass % (FFM%)	61.7	69.6	69.1	73.8	70.03	70.7
Total Body Water (TBW)	32.6	33.9	36.1	32.6	38.47	38.5
Total Body Water % (TBW%)	55.2	52.2	51.2	55	52.23	54.9
Hydration (TBW/FFM)	84.5	75.1	74.3	74.5	74.51	77.9
Extracellular Water (ECW)	21.8	18	18.4	16.7	18.87	20.9
Extracellular Water % (ECW%)	67.2	53.8	50.9	50.9	49.94	54.1
Intracellular Water % (ICW%)	32.8	46.2	49.1	49.1	50.07	45.9
Edema Index (ECW/TBW)	0.67	0.53	0.51	0.51	0.49	0.54
Body Cell Mass (BCM)	12.2	20.9	23.6	21.1	26.09	22.2
Extracellular Mass (ECM)	26.3	24.3	25.0	22.7	25.7	27.3
Malnutrition Index (ECM/BCM)	2.2	1.2	1,1	1.1	1.0	1.2
BCMI	4.9	8.1	8.8	8.5	9.27	9.1
Phase Angle (PhA) 50 kHz (°)	2.8	4.7	5.1	5.1	5.3	4.5
NA*_e_*:K*_e_* Ratio	1.8	1.3	1.1	1.2	1.16	1.3
Nutrition	370.7	628.7	695.1	639.2	754.0	689.9

Bioelectrical Impedance Analysis (BIA); Resistance (Rz); Reactance (Xc); Phase Angle (PhA); Fat-Free Mass (FFM); Total Body Water (TBW); Extracellular Water (ECW); Intracellular Water (ICW); Body Cell Mass (BCM); Fat Mass (FM); Fat-Free Mass (FFM); Na:K exchangeable rate (Na*_e_*:K*_e_*); Muscular Mass (MM); Body Cell Mass Index (BCMI).

**Table 2 ijms-25-09957-t002:** Patients’ characteristics and thyroid functional status.

Thyroid Function				Age (Years)	Gender (F:M)	BMI (w/h^2^)
	FT3 (pg/mL)	FT4 (ng/mL)	TSH (μU/mL)			
Hyperthyroid (n = 21)	5.4 ± 7.3	3.6 ± 8.4	0.1 ± 0.1	55.9 (35–84)	10.5:1	25.8 ± 5.0
Euthyroid (n = 57)	2.9 ± 0.4	1.2 ± 0.3	1.7 ± 0.8	54.8 (17–79)	7.1:1	26.9 ± 5.7
Hypothyroid (n = 25)	2.3 ± 0.7	1.0 ± 0.7	40.6 ± 53.6	56.4 (26–88)	9:1	26.2 ± 6.5
**Total** **(n = 103)**	**3.3 ± 3.3**	**1.6 ± 3.8**	**10.8 ± 31.1**	**55.4 (17–88)**	**7.5:1**	**26.5 ± 5.7**

Patients’ characteristics and their thyroid functional status. Mean values, ± standard deviation, for each group are reported. Minimum and maximum age for each group is indicated in brackets.

## Data Availability

The data that support the findings of this study are available from the corresponding author (S.S.), upon reasonable request.

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
