# Peer review of "Myxedema in Both Hyperthyroidism and Hypothyroidism: A Hormetic Response?"

_ijms, 2024, doi:10.3390/ijms25189957_

Round 1
Reviewer 1 Report
Comments and Suggestions for Authors
The article is interesting, and the outcomes are clearly presented.
Below my comments:
Abstract: "BIA offers no additional valuable information in evaluating fluid body composition in subclinical thyroid dysfunctions, but it represents a valuable method to analyze and monitor body fluid composition and distribution in severe thyroid dysfunctions" - rewrite it, please, the conclusion is arduous to understand.
Line 74: "particularly visible in two of them with hyperthyroidism lasting for more than three months" - mark the cases in Figure 1 as a different color of stars.
Figure 1: Remove multiple "1", please, or give the number of a particular case in the group (from 1 to n). B and D - give the unit names of the Y-axes.
Figure 1: Lines 80-82 - described axes in an ellipse (semi-minor and semi-major axes) or X and Y?
Figure 2 and Figure 3 contain repeated information - mark the significance in Figure 2 (for each pair: control [Euthyroid] vs. tested group [subsequent: Overt/Sub/Hyperthyroid]) and delete Fig. 3, please.
Discussion:
Lines 131–159 (BIA general information) and 245-262 (hormesis and hormone explanation): move it to the Introduction section, please.
Can BIA give a similar outcome as Diffusion-Weighted MRI for subcutaneous edema? Are there available other complementary methods available (giving insights into edema and cellularity)? What about the issue of the localized changes (not total-body changes of edema)?
Author Response
Comments:
- Abstract: "BIA offers no additional valuable information in evaluating fluid body composition in subclinical thyroid dysfunctions, but it represents a valuable method to analyze and monitor body fluid composition and distribution in severe thyroid dysfunctions" - rewrite it, please, the conclusion is arduous to understand.
Response
We have clarified the meaning of the phrase by adding the word “overt” to better explain the concept. We intended to say that, according to the results of our study, BIA has no utility in “subclinical” thyroid disorders but it is indeed very useful in “overt” and “severe” thyroid dysfunctions.
- Line 74: "particularly visible in two of them with hyperthyroidism lasting for more than three months" - mark the cases in Figure 1 as a different color of stars.
Response
We have marked the two cases by adding two black arrows in panel C of Figure 1. The addition of the two black arrows in indicated also in the legend of the figure
- Figure 1: Remove multiple "1", please, or give the number of a particular case in the group (from 1 to n). B and D - give the unit names of the Y-axes.
Response
Unfortunately, we are unable to eliminate multiple “1” in Figure 1. The number 1 is given by the program and it refers to the first measurement for each patient. The name of the Y axes in all panels is Xc/H and in brackets we have also reported the unit names (W/m)
- Figure 1: Lines 80-82 - described axes in an ellipse (semi-minor and semi-major axes) or X and Y?
Response
The two axes are those of the ellipse and not to the X and Y axes. We have changed their description in the legend of Figure 1 into “semi-major axis of the ellipse” and “semi-minor axis of the ellipse” respectively
- Figure 2 and Figure 3 contain repeated information - mark the significance in Figure 2 (for each pair: control [Euthyroid] vs. tested group [subsequent: Overt/Sub/Hyperthyroid]) and delete Fig. 3, please.
Response
We accept the suggestion of the reviewer and we thank him/her because we realized that the results are now more clearly presented, without any duplication. As suggested by the Reviewer, we have removed Figure 3 and we have included the significance in Figure 2. Any citation of Figure 3 in the text has been removed and changed into Figure 2.
Comments - Discussion:
- Lines 131–159 (BIA general information) and 245-262 (hormesis and hormone explanation): move it to the Introduction section, please.
Response
Following the reviewer’ suggestion we moved the paragraph concerning the BIA general information and Hormesis and Hormones explanation from the Discussion to the Introduction. The references were modified accordingly.
- Can BIA give a similar outcome as Diffusion-Weighted MRI for subcutaneous edema? Are there available other complementary methods available (giving insights into edema and cellularity)? What about the issue of the localized changes (not total-body changes of edema)?
Response
We than the reviewer for the suggestions regarding the possible use of Diffusion-Weighted MRI in the evaluation of oedema. We added a comment and some references (7 new references, # 48-54) regarding the use of this technique to detect and measure oedema in different clinical conditions, including the possibility to use it in Grave’s ophthalmopathy. The reference section has been modified and references re-numbered accordingly and the new references have been highlighted in yellow. However, we couldn’t find any study in the literature concerning the use of this technique in the evaluation of generalized oedema in both hypothyroidism and hyperthyroidism. Moreover, the application of this technique in the course of routine out-patients visits appears difficult to be performed. Its application is, in fact, limited by the cost and the technical difficulties that required trained and expert staffs. We have added a comment about that at the end of the discussion section.
We thank again the reviewer for the useful suggestion. In our study we have used the BIA 101 analyzer apparatus, that allows total-body changes. The use of other BIA analyzers, able to perform local measurements, could be useful to evaluate local alterations of fluid distribution, such as the pretibial myxoedema observed in patients suffering from Graves' disease. We are planning to use the new BIA 101 BIVA® PRO by Akern, to perform such regional BIA measurements.
Reviewer 2 Report
Comments and Suggestions for Authors
The authors should refer in detail to the pathophysiology of myxedema in both hypothyroidism and hyperthyroidism.
The authors should try to explain what they mean by a hormetic response.
Comments on the Quality of English LanguageMinimal editing of English required.
Author Response
Comments:
- The authors should refer in detail to the pathophysiology of myxedema in both hypothyroidism and hyperthyroidism.
Response
We reported in the Introduction section that the exact pathogenic mechanism of myxoedema occurring in thyroid disorders is not clear yet and the reason why it can be observed in both hypothyroid and hyperthyroid patients is not known either.
According to pathogenic studies reported in the literature, often very old and based on histochemical studies, the condition of mixoedema is due to the accumulation of acid mucopolysaccharides and probably also by inadequate drainage of the lymph (see paper by H.H. Parvin et al New Engl Med 1979, 301 (9),460-5). Other evidences indicate that there is also a concomitant expansion in total body water and sodium and a possible role of the anti-diuretic hormone has been suggested (see Pettinger WA, et al. Inappropriate secretion of anti-diuretic hormone due to myxedema. N Engl J Med. 1965; 272, 362–364).
- The authors should try to explain what they mean by a hormetic response.
Response
We reported the following phrase in the Introduction section to try to explain the concept of hormetic response: “Hormetic response in medicine represents and adaptive response of the cells, tissues, organs and organisms to changes in environmental conditions and to exposure to several different stressors, such as metals, vitamins, macronutrients, micronutrients, drugs and, virtually, to every physical, chemical and potentially toxic agent.” The principle of hormesis is based on the concept that what makes a substance lethal or toxic is related to its dosage. Many toxic substances including iron, selenium and even common elements as water and oxygen, are used by our cells and tissue and represent necessary components for the normal function of our body. However, when they are present in the body at both low or high concentrations, they could be responsible for damages and may even lead to death of the cells and of the tissues and, consequently, of the subject.
In our study, we postulated that the same kind of adaptation observed with nutrients and stressors could be valid also with respect to the exposure of the body cells to the effects of thyroid hormones. According to the hormesis and considering the U-Shaped or the inverted U-shaped curves of BIA parameters observed in our patients with both hypothyroidism and hyperthyroidism, we concluded that either severe hypothyroidism and overt hyperthyroidism are responsible for the induction of myxoedema in a typical hormetic-type response.
Round 2
Reviewer 2 Report
Comments and Suggestions for Authors
The manuscript has been sufficiently improved.
Comments on the Quality of English LanguageMinor editing of English language required.